# The Fetal Environment and the Development of Hypertension—The Epigenetic Modification by Glucocorticoids

**DOI:** 10.3390/ijms26010420

**Published:** 2025-01-06

**Authors:** Fumiko-Kawakami Mori, Tatsuo Shimosawa

**Affiliations:** 1Department of Endocrinology, Mitsui Memorial Hospital, Tokyo 101-8643, Japan; 2Department of Clinical Laboratory, School of Medicine, International University of Health and Welfare, Otawara 324-8501, Japan; tshimo-tky@umin.ac.jp

**Keywords:** developmental origins of health and disease (DOHaD), environment, DNA methylation, epigenetics, blood pressure, hypothalamus

## Abstract

Intrauterine growth restriction (IUGR) is a risk factor for postnatal cardiovascular, metabolic, and psychiatric disorders. In most IUGR models, placental dysfunction that causes reduced 11β-hydroxysteroid dehydrogenase 2 (11βHSD2) activity, which degrades glucocorticoids (GCs) in the placenta, resulting in fetal GC overexposure. This overexposure to GCs continues to affect not only intrauterine fetal development itself, but also the metabolic status and neural activity in adulthood through epigenetic changes such as microRNA change, histone modification, and DNA methylation. We have shown that the IUGR model induced DNA hypomethylation in the paraventricular nucleus (PVN) in the brain, which in turn activates sympathetic activities, the renin–angiotensin system (RAS), contributing to the development of salt-sensitive hypertension. Even in adulthood, strong stress and/or exogenous steroids have been shown to induce epigenetic changes in the brain. Furthermore, DNA hypomethylation in the PVN is also observed in other hypertensive rat models, which suggests that it contributes significantly to the origins of elevated blood pressure. These findings suggest that if we can alter epigenetic changes in the brain, we can treat or prevent hypertension.

## 1. Introduction

In 1989, Barker et al. reported that low birth weight is associated with hypertension and cardiovascular disease later in life [1,2]. They showed that epidemiological studies in areas with high neonatal mortality in the 1920s also showed high cardiovascular disease mortality in the 1970s. This epidemiological study showed an increased cardiovascular risk in low-birth-weight infants weighing less than 2500 g. During the Dutch famine of World War II (1944–1945), pregnant women were exposed to a low-nutrition diet of less than 1000 kcal/day, fetuses were born as low-birth-weight babies and had not only a higher risk of developing hypertension, diabetes, and other known cardiovascular diseases, but also psychiatric disorders such as autism and schizophrenia in adulthood.

The developmental origins of health and disease (DOHaD) hypothesis was proposed by Gluckman, Hanson, and their colleagues because growth restriction due to undernutrition alone cannot explain long-term metabolic and neurological changes [3]. A preceding cohort study on famine in the Netherlands, which assessed the DNA methylation of whole blood cells, reported the persistence of epigenetic alterations in subsequent years [4,5]. It is also clear that changes affect subsequent generations [6]; however, how these changes occur and are propagated from one generation to the next and beyond is not fully understood. It is well accepted that gene expression patterns are regulated by epigenomic adjustments besides gene mutations, and among epigenomic modifications, DNA methylation is reversible and well controlled. In fertilized eggs, DNA methylation is reset to a demethylated state, and therefore, maternal DNA methylation patterns are not necessarily inherited by the next generation. One reason why dysregulated DNA methylation could be transferred to future generations may be that infants who are born as IUGRs are exposed to a similar intrauterine and placental blood flow environment when they are conceived. For example, changes in the hypothalamic–pituitary–adrenal (HPA) axis and insulin resistance, especially in low-birth-weight infants, may continue to be influenced by the mother’s fetal environment and affect the child’s subsequent weight gain and stress tolerance. Fetal nutritional status also affects histone modifications, and it has been reported that the histone acetylation status around some genes is changed [7,8]. Histone modifications are also known to affect the methylation status of DNA.

In addition, there is evidence suggesting that microRNAs in the father’s sperm and the mother’s body may influence fetal development after fertilization. Indeed, paternal obesity is thought to affect offspring through microRNA-mediated mechanisms [9]. Interestingly, prenatal GC overexposure has also been reported to alter the microRNA profile of sperm in offspring, which is maintained in the next generation [10,11].

DNA methylation is primarily modified during the fetal stage, with much of it being retained thereafter. Hence, the fetal environments are critical for epigenetic modifications.

There are three types of regulation of gene expression: regulation by microRNAs, histone modifications, and DNA methylation. The duration of each modification differs, and DNA methylation modification is considered to be the most long-lasting of them all. Although microRNA modifications are generally short-lived, they are thought to fine-tune gene expression because a single microRNA change can modify the expression of many genes, and conversely, multiple microRNAs can target the same gene. This microRNA can also indirectly modify DNA methylation by regulating the expression of both DNA methyltransferases and demethyltransferases [12,13,14]. Together with histone proteins, DNA forms a complex called chromatin, and conformational changes in this chromatin regulate the ease of binding of transcription factors and nuclear receptors, as well as transcriptional activity. Histones undergo a series of modifications, including acetylation, methylation, and phosphorylation, which alter their chromatin state and regulate gene transcription. Histone modification is also closely related to the methylation status of DNA. With respect to histone modification and DNA methylation regulation, the trimethylation of lysine 9 on histone H3 protein subunits (H3K9me3)-dependent heterochromatin plays a role in DNA methylation reprogramming in the postimplantation embryo [15]. This H3K9me3 change has not been observed in sperm or eggs in the IUGR model and may not be affected by the fetal environment [16]. Conversely, the nutritional status impacts histone acetylation as previously noted [7,8], and it has been shown that HDAC inhibitors can enhance the HPA axis, which is disrupted by fetal stress. This suggests that the fetal environment also exerts a direct influence on histone modifications [17].

Consequently, a number of steps regulate the reading of genetic information.

Among the fetal environment, in addition to progesterone, GCs are considered to have a very large impact on epigenetic modifications. Low nutrition, stress, older childbearing, and smoking, as discussed below, result in excessive fetal exposure to GCs. We will discuss GCs and DNA modification and the mechanisms that are crucial to the blood pressure center.

Furthermore, excess GC states strongly affect the epigenetic modification and have been shown to be a major cause of IUGR, and this review discusses GC regulation and epigenetic modifications.

## 2. Roles of GCs in Pregnancy and Their Regulation in the Fetus

During pregnancy, the maternal total cortisol concentration increases up to 4–20-fold [15,18]. As cortisol production increases, so too does the corticosteroid-binding protein, so that free cortisol concentrations remain relatively constant until the third trimester [19,20]. In late pregnancy, free cortisol levels increase dramatically. This elevated cortisol level is necessary for maternal immune regulation and fetal maturation. Cortisol, together with progesterone, regulates regulatory T cells and is thought to be involved in the immune tolerance of the mother [21]; the balance between cortisol and progesterone signaling is also important for the continuation of normal pregnancy. Cortisol also plays an important role in placental angiogenesis and the maturation of fetal tissues, such as the lungs, kidneys, and brain. Therefore, the roles of GCs in pregnancy are diverse [22,23,24].

The placenta controls the nutritional, hormonal, and immune responses that are transferred from the mother to the fetus and supports fetal development. Maternal cortisol passes through the placenta, which expresses high levels of 11 hydroxysteroid dehydrogenase 2 (11βHSD2, which converts cortisol to inactive form, cortisone) protecting the fetus from cortisol overexposure. Regarding fetal cortisol production, the adrenal glands start to develop in the first trimester in an adrenocorticotropic hormone (ACTH)-independent manner. In the human fetus, later, the adrenal glands develop due to ACTH and fetus-derived growth factors from 15 weeks of gestation. Maternal ACTH does not pass through the placenta, although maternal cortisol patterns affect fetal circadian rhythm [25]. It is hypothesized that the fetus receives information dependent on the external environment (maternal circadian rhythm, nutrition, and stress conditions) from the mother, as well as endogenous control of fetus.

## 3. Animal Models That Result in GC Overexposure in the Placenta

### 3.1. Oxidative Stress and 11βHSDs

As mentioned above, the fetus is protected from excess maternal cortisol by placental 11βHSD2 and fetal 11βHSD2. The metabolism of cortisol involves two isoforms of 11βHSD, namely 11βHSD1 and 2; 11 βHSD1 converts inactive cortisone to active cortisol in an NADPH-dependent manner, and 11βHSD2 converts cortisol to cortisone, an biologically inactive form of cortisol, in an NAD^+^-dependent manner. Thus, local cortisol levels are regulated by redox signals. In a study examining the relationship between 11βHSD2 expression and inflammatory cytokines in the placenta, the expression of nuclear factor-kappa B, interleukin (IL)-1β, and IL-8 was elevated in the group with decreased 11βHSD2 expression [26]. Of note, inflammation not only causes a decrease in 11βHSD2 activity, but may also be involved in the decreased expression of 11βHSD2.

### 3.2. GC and Its Receptors

Cortisol receptors mainly comprise GRs and mineralocorticoid receptors (MRs). Cortisol also binds to the progesterone receptor (PR), albeit with lower affinity. The binding affinity for the MRs is equal in cortisol and aldosterone; however, cortisol concentrations are 1000-fold higher than aldosterone concentrations. Hence, in parts of the tissue where 11βHSD2 is not expressed, cortisol is the primary ligand for MR. These two receptors cooperatively play an important role in the regulation of fetal development and organ function, and the regulation of local GC by 11βHSD2 is important. On the fetal side, 11βHSD2 is expressed in some tissues, such as the kidney, brain, and gonads, and this expression is also more widespread than in adulthood [27]. This could play a role in preventing GR overactivation and, at the same time, protecting the MR activation by cortisol. However, the excess cortisol can directly affect the tissues where 11βHSD2 are not expressed in the fetus, and placental 11βHSD2 acts as a gatekeeper to prevent cortisol overexposure in the fetus [28]. Although GC signaling is required for fetal lung and heart maturation and neurodevelopment, excess cortisol is known to cause IUGR, post-developmental obesity, and increased cardiovascular risk.

Interestingly, in an examination of brain-specific 11βHSD2 knockout (KO) mice, Wryrwoll et al. reported that IUGR was not present and the HPA axis was not disrupted [29,30]. Instead, this mouse presented depression, memory deficits, and decreased 5-hydroxytryptamine1A receptor expression, which is a serotonin receptor and its dysfunction is known to cause anxiety and memory loss. It is possible that the limited expression of 11βHSD2 in the brain may contribute to the limited change in the phenotype.

### 3.3. 11βHSD2 Activity and IUGR

As previously discussed, a decrease in 11βHSD2 activity in the placenta affects the fetal GC environment. The administration of dexamethasone (Dex), a synthetic GC that is not degraded by 11βHSD2, also causes excess GC.

The expression and activity of 11βHSD2 in the placenta have been studied in a variety of animal models and in humans. In aging rats, 11βHSD2 expression in the placenta of male fetuses decreased its expression, whereas it was rather increased in female fetuses, and though oxidative stress is increased in both fetuses, activity is expected to be lower in both [31]. Heavy smoking results in cadmium exposure, and cadmium reduces both 11βHSD2 expression and activity in the placenta [32]. As for the alcohol exposure model, although some results are inconsistent across studies, prenatal alcohol exposure seems to also increase the concentration of corticosteroids in the fetus via the inhibition of 11βHSD2 activity [33,34,35]. Maternal psychological stress also decreases 11βHSD2 expression and activity in the placenta, and the fetus is overexposed to GCs [36,37,38,39]. Maternal alcohol consumption, smoking, aging, and psychological stress have been reported to reduce 11βHSD2 mRNA expression in the placenta [32,40,41]. We also observed reduced expression of 11βHSD2 in the placenta of a low-protein (LP) diet model of undernutrition in Sprague Dawley rats [42]. In the IUGR model, obesity, diabetes, hypertension, and an increased risk of cardiovascular disease occur in addition to neurodevelopmental impairments, such as autism and anxiety, as shown in Table 1. These phenotypes caused by exposure to stress and low nutrition in the fetal environment.

## 4. Changes in Each Organ in IUGR Model and Their Effects on Blood Pressure

Glucocorticoid overexposure in the placenta causes postdevelopmental abnormalities by altering metabolic signaling set points, along with developmental abnormalities in many tissues.

Among these phenotypes, hypertension is caused by various factors. Blood pressure is regulated by the autonomic nervous system and humoral factors, and its target tissues are heart, kidney, and blood vessels.

In addition to the effects on the sympathetic nervous system, there are also reports that IUGR directly affects organs and causes hypertension. The effects on each organ are described in turn below.

### 4.1. Kidney

In the IUGR model, several organ abnormalities have been reported as the cause of hypertension, one of which is the kidney [58,59,60]. Nephrogenesis takes place in the second to mid-third trimester of pregnancy, with the formation of the nephrons being complete by 32 to 37 weeks of gestation in humans, and the nephron number is reduced in the IUGR fetus. Although the reduced number of nephrons is thought to be a developmental disorder rather than an epigenetic change, which will be discussed further on, the impairment of the nephron number can lead to chronic kidney disease, which in turn predisposes to salt-sensitive hypertension. In addition to this developmental abnormality, 11βHSD2 mRNA expression is decreased and it causes hypertension via mineralocorticoid receptor activation. This reduced 11βHSD2 expression is achieved by a reduction in the transcription factor, hyper methylation of the promotor region of the 11βHSD2 gene, and histone tri-methylation [61]. In addition, by gaining weight, perirenal and renal sinus fat stimulate renin secretion by depressing the arterioles in the kidneys, which activates the RAS. Adipocytokine released from adipocytes stimulates aldosterone release via ERK1/2 and Wnt pathways, as well as activating MR signal transduction [62,63]. It should be noted that the change does not occur directly in the kidneys; rather, sympathetic activation plays a role in both vasoconstriction and sodium retention via Na-Cl cotransporters in the distal tubules, which ultimately leads to an elevation in blood pressure [64].

### 4.2. Blood Vessels and Heart

It has been reported that intima media are thickened, and impaired endothelium-dependent vasodilation in low-birth-weight infants [65,66]. In the IUGR model on a low-protein diet, endothelium-dependent vasorelaxation was impaired from a young stage before blood pressure increases, and as the weeks of age progressed, there was an increased expression of L-arginase and induced uncoupling of e-NOS, which are components involved in vasorelaxation, along with increased blood pressure [67]. In addition, an examination of fetuses in the folate deficiency model, a methylated donor, showed progressive remodeling in aorta after aging [56].

The incidence of cardiac disease is increased in with IUGR as a consequence of hypertension, but there is also a direct effect on the myocardium itself, independent of blood pressure. Cardiomyocytes of IUGR fetuses are small in size, immature, and exhibit reduced cell cycle activity [68]. It has also been reported that IUGR causes abnormalities in myocardial mitochondrial function, and changes reactive oxygen stress-related genes [69,70]. Further investigation is needed to determine how the GR signal and epigenomic modifications affect changes in blood vessels and the heart.

### 4.3. Adipose Tissue

The IUGR model produces postdevelopmental obesity, and obesity is a major contributor to hypertension. However, in addition to increased circulating blood volume and peripheral vascular resistance, obesity related hypertension is induced by various factors. One of the causes is leptin produced from adipocyte, which is increased from birth in the IUGR model. Leptin exerts its effects both centrally and peripherally, as a hormone that suppresses appetite, as a cytokine that induces the production of inflammatory cytokines, including TNFα, IL-6, and IL-12 [71]. Inflammatory cytokines induce the fibrosis and hypertrophy of the target tissues such as heart and blood vessels, insulin resistance, and the activation of the RAS and sympathetic nervous system [72,73,74]. Leptin signaling stimulates the PVN and activates sympathetic nerves via the stimulation of pro-opiomelanocortin (POMC) neurons in the arcuate nucleus (ARC).

### 4.4. Brain

In the IUGR model, the hyperactivity of the sympathetic nervous system occurs. This has been shown to be due to multiple causes, including the enhancement of the HPA axis, increased leptin concentration, and enhancement of the RAS [74]. As mentioned above, increased leptin production from adipocytes causes increased sympathetic activity via the activation of P neurons in the ARC of the brain. The network system of sympathetic nervous system regulation in the brain and sympathetic activation by the RAS are discussed below.

Among these, the change in the central nervous system, which coordinates the tone of all tissues, is thought to play a fundamental role in disease development, and furthermore, given the plasticity of the neuron, it may be a better target for epigenomic transformation than other tissues.

## 5. IUGR, GC Overexposure and Epigenetic Change

Epigenetic modifications of gene expression by GC signaling have been shown to have a wide range of effects, including DNA methylation, histone modifications and the induction of microRNAs, and the sites of action are systemic, including those observed in central nervous system development.

In the IUGR model, whether it is a direct effect of GCs is not clear, though increased histone 3 lysine 9 and histone 3 lysine 14 are observed along with decreased expression of histone deacetylase (HDAC)1 in the brain [75]. GR, together with cofactors, recruit HDAC2 and regulate the inflammatory gene expression [76]. It has been reported that HDAC2, along with HDAC1, is upregulated in nicotine-induced IUGR models and influences cartilage development [77]. It is also interesting to note that HDAC inhibitors are attracting attention as a treatment for hypertension [78]. In contrast, microRNAs play a role in the fine-tuning of signals. For example, miR92a-3p, in conjunction with PTEN and AKT, regulates the expression of GC downstream signaling [79]. Additionally, histone modifications and microRNAs play a role in the susceptibility to GR signaling. Redox signals mediate HDAC and modulate GR signaling [80]. Some microRNAs such as miR29, miR124a, miR86-5p, and miR148a lead to resistance to GCs signals [80,81,82,83].

In neural progenitor cells, Dex treatment induces global hypomethylation, which is accompanied by a decrease in DNA methyltransferase 3a (DNMT3a) mRNA expression. The downregulation of Dnmt3a was dependent on ten-eleven translocation methylcytosine dioxygenase 3 (Tet3) overexpression and persisted after cell passaging. This is also observed in the cortex of mouse pups exposed to Dex in utero [84].

It is examined that GR over-activation can cause methylation changes in adulthood. These include the demethylation of CpGs around the GC responsive element, which is facilitated by the activation of the Tet family and the reduction in DNMT expression. However, the duration of the changes is shorter than in the case of fetal GC exposure. It is possibly because neurons are actively dividing during development and structural changes may occur. Further research is needed to elucidate the differences in epigenomic modifications during the so-called critical period [85,86].

Excess GCs alter the methylation of many genes involved in metabolism and neural activity, which can be maintained throughout adult life. Even in adulthood, DNMT3a and DNMT1 play pivotal roles in neural activity. Moreover, the conditional knockout of DNMT1 and DNMT3a in the forebrain and post-mitotic neurons impairs memory formation and its maintenance [87].

In our study, Dex loading and a low-protein diet showed a decrease in DNMT1 and DNMT3a in the PVN both at postnatal day 7, and this decrease was maintained in adulthood, although no difference was found for the Tet family. This epigenomic change altered the expression of many genes in the PVN of Dex-treated rats, and a gene ontology pathway analysis showed elevated gene expression associated with the calcium signaling pathway, serotonergic synapse, dopaminergic synapse, oxytocin signaling pathway, and aldosterone synthetic pathway, among others [42]; these pathways can affect central sympathetic nervous system activation and the RAS.

Therefore, the regulation of DNA methylation affects neural activity not only during neurodevelopment but also after development.

The HPA axis was also altered in the GC exposure model, suggesting that postdevelopmental GC changes may also affect the maintenance of DNA methylation. Our model also revealed increased CRH mRNA levels in the PVN after development.

## 6. Blood Pressure Regulation in the Brain

### 6.1. The Central Nervous Systems Which Regulate Sympathetic Nerve Activity

The rostral ventrolateral medulla (RVLM), a nucleus in the medulla oblongata, projects to the sympathetic nervous system, and its activation causes sympathetic hyperactivity. The RVLM receives inputs from the PVN and caudal ventrolateral medulla (CVLM). In the RVLM, the MR and the angiotensin receptor type 1a receptor (AgtR1a) are present and the cortisol, aldosterone, and angiotensin II can directly activate the RVLM.

The PVN is a nucleus that integrates some inputs from multiple nuclei as follows [88].

Interoceptive pathways, including nociceptors, baroreceptors and thermo receptors, are projected to the PVN via CVLM and nucleus of tractus solitarius. This input from each organ’s information allows the PVN to regulate sympathetic activity to maintain homeostasis.

Inputs from the circumventricular organs, such as the subfornical organ and organum vasuculosum of the lamina terminalis, which are located around the third ventricle, are not affected by blood–brain barrier, and contain salt-sensing neurons, regulate antidiuretic hormone secretion and sympathetic nerve activity by sensing salt and water intake in the body.

Inputs from the limbic system, such as the prefrontal cortex and amygdala via the bed nucleus and dorsomedial hypothalamus, are related to stress response.

The limbic system also receives projections from POMC neurons and neuropeptide-Y neurons in the ARC, the appetite center, and obesity-induced sympathetic nervous system hyperactivity is mediated by POMC-PVN activation [89].

Further, PVN directly receives humoral signals such as GCs, glucose, and angiotensin II.

The activation of PVN neurons can cause an increase in sympathetic activity, the HPA axis, and vasopressin (VP) secretion by signaling to downstream nuclei.

### 6.2. Angiotensin II Signal in the PVN

Previous reports have shown that interference with the AgtR1a in PVN neurons reverses salt-sensitive hypertension in mRen2 transgenic rats [90]. De Kloet et al. reported that the optical stimulation of AgtR1a-expressing neurons in the PVN using the adeno-associated virus gene transfer of enhanced yellow fluorescent protein and a light-sensitive ion channel induced blood pressure elevation [91].

Our study demonstrated salt-sensitive hypertension and the increase in the mRNA expression of AgtR1a in the PVN in both Dex-induced and LP diet-induced IUGR models. These models were accompanied by the reduced expression of DNMT1 and 3a and reduced methylation in the promoter region of AgTR1a in the PVN (Figure 1).

### 6.3. The Role of DNA Methylation in the Brain for Blood Pressure Regulation

To investigate the causes of demethylation of the promoter region of the AgtR1a in these models, we performed a ChIP assay for DNMT1 and DNMT3a, and found decreased the binding of DNMT3a but not DNMT1. In our study, the direct GC treatment of cultured hypothalamic cells reproduced the decreased expression of DNMTs and decreased the binding of DNMT3a to AgtR1a, which is consistent with the effects of GC on neural progenitor cells as previously described. These results suggested that the decrease in DNMT3a expression induced by GC contributed to the increase in AgtR1a expression (Figure 1). It is not clear whether changes in AgtR1a really contribute to salt-sensitive hypertension in the IUGR model, since other signaling modulators of the PVN are known, including CRH and the RAS. Therefore, Dex was administered to pregnant AgtR1a KO mice to create an IUGR model. Fetuses of Dex-treated AgtR1a KO mice developed IUGR and subsequently became obese but did not develop salt-sensitive hypertension. Furthermore, among Sim1-Cre/Dnmt3aKO mice, hypothalamus-specific Dnmt3aKO mice showed IUGR and post-developmental salt-sensitive hypertension, and increased expression of AgtR1a in neurons in the PVN (Figure 1). These findings suggested that fetal overexposure to GCs by Dex administration or an LP diet results in increased RAS signaling and salt-sensitive hypertension in the PVN via a decrease in Dnmt3a.

Another study on hypothalamus-specific Dnmt3aKO mice showed that they developed obesity, lipid abnormalities, insulin resistance, and hypertension, similarly to the IUGR model [92]. More interestingly, in spontaneously hypertensive rats (SHRs)—a model of hypertension generated by selective breeding of rats for blood pressure—AgtR1a expression in the PVN increased with Na-K-Cl cotranspotor1 and was accompanied by decreased binding of DNMT1 and DNMT3a to AgtR1a. In addition, an injection of a DNMT inhibitor to the PVN of WKY rats, a non-hypertensive control group, caused an increase in AgtR1a expression and blood pressure, indicating a direct effect of DNMT on blood pressure in the PVN [93]. Unlike the IUGR model, the expression of DNMT1 and DNMT3a itself is not reduced in SHRs, and it is interesting that it is not yet clear what mechanism reduces the binding of DNA methyltransferases to genes associated with blood pressure.

These results indicate that in most of the IUGR models, the excessive activation of GR during the fetal period decreases the expression of DNMTs in the PVN and increases the expression of AgTR1a, resulting in salt-sensitive hypertension. And that the decrease in the binding of DNMTs to AgTR1a in the PVN plays a major role in the development of hypertension in other hypertension models that do not cause IUGR, and may be a relatively universal event in the development of hypertension (Figure 2).

### 6.4. Aging and Environmental Effect on DNA Methylation

As mentioned earlier, methyltransferases and histone modifications such as DNMT1,3a act on postmitotic neurons in the brain and thus contribute to the regulation of neuronal activity in adult life [87]. However, it is not clear whether the regulation of methylation in all regions of the brain is regulated to the same extent by excess fetal GC. Phenotypically, depression, anxiety, and sympathetic activation have been observed, and no abnormalities in higher functions have been reported, suggesting that the regulation differs from region to region of the brain.

In recent years, it has become possible to assess DNA methylation status in individual cells, and it is now known that the degree of methylation varies by region and cell type in the brain [94,95]. Using this information, it will be possible to determine which regions of the brain are more epigenetically altered by overexposure and which regions are susceptible to fluctuations.

With regard to DNMTs in the brain, they are known to decrease with age, which not only make it difficult to coordinate the fluctuations in activity in neurons to acquire or maintain memory, it can also result in hypertension due to sympathetic activation, as we have shown here.

## 7. Future Perspectives to Reset the Environmental Effect of Fetus

The fetal period is a critical period that determines metabolic and psychological set-points, and epigenetic modifications are deeply involved. This is the mechanism by which the child adapts to environmental changes such as stress and undernutrition, but the permanence of these changes results in postdevelopmental incompatibilities. One of these changes was found to alter gene expression in the hypothalamic PVN, resulting in salt-sensitive hypertension.

The persistence of these effects is a social problem, but considerable research has been conducted to ameliorate them, including folic acid supplementation, IGF-1 treatment, and exercise [50,96,97,98,99,100]. Epigenetic modifications are highly plastic in the early postnatal period.

It has been reported that decline in DNMTs can be prevented to some extent by exercise [101]. If DNMTs can be activated in the brain, it may be possible to improve the fetal effect, including blood pressure regulation. On the other hand, DNMTs are activated in tumor cells [12,102], and considering that DNMT inhibitors are used as anti-tumor drugs, it may be difficult to create a drug that moderately activates DNMTs in a tissue-specific manner, but this is an area where future developments in therapeutic technology are to be expected.

Therefore, if an intervention cannot be performed during pregnancy, it is advisable to do so soon after birth. Further development of safe intervention methods for epigenetic modifications is required.

## Figures and Tables

**Figure 1 ijms-26-00420-f001:**
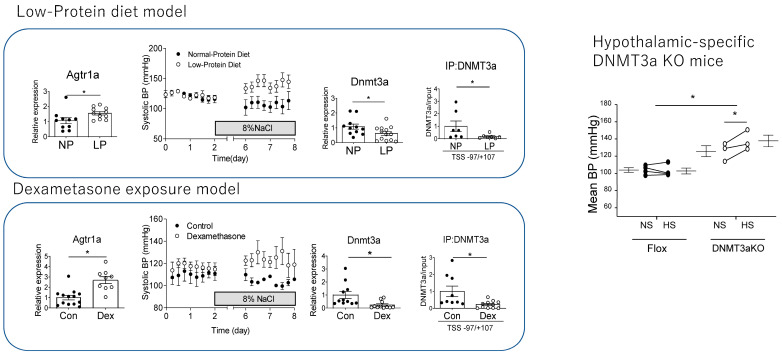
Low-protein diet and Dex treatment decreased mRNA expression of Dnmt3a, binding promotor region of Agtr1a, and mRNA expression of increased AgtR1a. Both models developed salt-sensitive hypertension. Hypothalamic-specific Dnnmt3a KO mice also developed salt-sensitive hypertension (* represents significant difference *p* < 0.05 on unpaired *t*-test). Modified from authors’ data [31].

**Figure 2 ijms-26-00420-f002:**
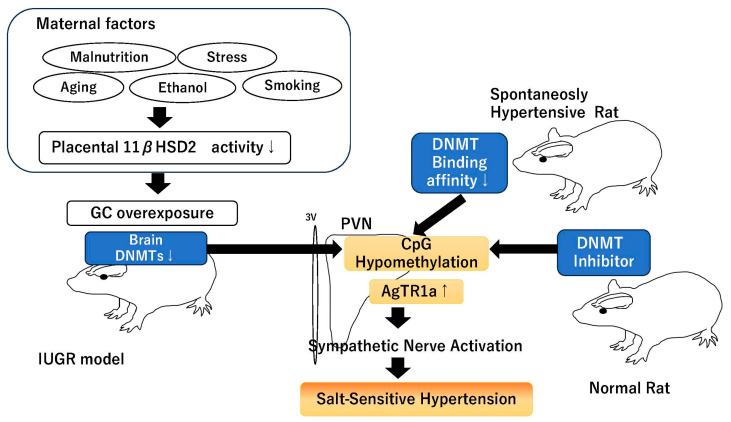
Scheme of `Role of DNMTs in PVN on salt-sensitive hypertension’. ↓ indicates decreased activity or expression, ↑ indicates increased expression.

**Table 1 ijms-26-00420-t001:** Rodent models of IUGR and phenotype.

Model	Target Tissue	Phenotype	
**Dexamethasone exposure**	Heart	Heart rate, cardiovascular dysfunction	[43]
	Heart, brain, kidney, blood vessel	Salt-sensitive hypertension	[42,44]
	Kidney	Chronic kidney disease	[44,45,46]
	Brain	Anxiety, depression	[47,48]
	Adipose tissue	Obesity	[49]
**Low-protein diet**	Heart	Cardiomyocyte change Cardiac dysfunction	[50,51,52]
	Heart, brain, kidney, blood vessel	Salt-sensitive hypertension	[42]
	Adipose tissue	Obesity	[53,54]
**Ethanol exposure**	Brain	HPA axis abnormality, depression	[33,41]
**Chronic stress**	Brain	HPA axis abnormality, neurogenesis	[38]
	Heart, brain, kidney, blood vessel	Salt-sensitive hypertension	[55]
**Folic acid deficiency**	Blood vessel	Remodeling of aorta, Hypertension Body weight	[56,57]

## Data Availability

The authors permit to share all the data.

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
