# Peer review of "The Fetal Environment and the Development of Hypertension—The Epigenetic Modification by Glucocorticoids"

_ijms, 2025, doi:10.3390/ijms26010420_

Round 1
Reviewer 1 Report
Comments and Suggestions for Authors
In the manuscript titled “Central blood pressure regulation and epigenetics – From prenatal environmental to adulthood,” the authors review the literature surrounding the effects of glucocorticoids on development from fetal to adult.
Comments and Suggestions
The current title is broader than the topic that is being reviewed. For example, epigenetic discussion is mostly limited to miRNAs and DNA methylation, glossing over histone modifications and transcriptional effects from changes in cis-regulatory elements.
More references in the introduction would strengthen the background of the paper
Author Response
Thank you very much for your peer review.
The title has been rewritten to be closer to what was reviewed.
As for histone modification, we have added a citation of a paper related to DoHAD and histone modification in the Introduction section(yellow highlighted area in 6-3.).
However, there are no reports related to DoHAD and the sympathetic nervous system, so we have not added them to the respective sections.
Reviewer 2 Report
Comments and Suggestions for Authors
Dear Authors.
The manuscript you submitted is an excellent piece of science. It is a meticulous review with many molecular details that give the reader an idea of how epigenetic modifications change tissue function and physiological responses to glucocorticoid overexposure.
However, the title of the manuscript points to the central regulation of blood pressure. After reading your paper, the central idea is that it deals with animal models with IUGR or those exposed to alterations in glucocorticoid metabolism and quantity. While the effect of glucocorticoids on blood pressure is mentioned (from section 4 onwards, out of seven total sections) and developed further in section 6, the idea remains that the vast majority of the review is about glucocorticoids and fetal overexposure to them.
They can take two paths: 1.- change the title, making a better reflection of overall manuscript, or 2.- modify some paragraphs of the document, emphasizing the effect of fetal overexposure to glucocorticoids and the effect that fetal programming of overexposure to CG has on the regulation of blood pressure, whether mediated by the SNS, regulation of peripheral vascular resistance or control of cardiac activity.
The work done is of outstanding quality and of great scientific soundness, but the authors should better specify the effect of CG on the programming of the mechanisms that regulate central blood pressure.
I reconsider the manuscript with major revisions.
Author Response
Thank you very much for your peer review.
As you pointed out, the first half of the article is mostly about glucocorticoid and DOHAD, so the title and the content of the article were far apart.We have decided to change the title because the content of glucocorticoids is also an important topic and we would like to include it as it is.
We have previously shown the effect of direct translation on GC and PVN by glucocorticoid administration to cultured hypothalamic cells showing AgTR1a expression, which we have added (yellow highlighted area in 6-3.).